# Ribosome-Directed Therapies in Cancer

**DOI:** 10.3390/biomedicines10092088

**Published:** 2022-08-26

**Authors:** Gazmend Temaj, Silvia Chichiarelli, Margherita Eufemi, Fabio Altieri, Rifat Hadziselimovic, Ammad Ahmad Farooqi, Ilhan Yaylim, Luciano Saso

**Affiliations:** 1Faculty of Pharmacy, College UBT, 10000 Prishtina, Kosovo; 2Department of Biochemical Sciences “A.Rossi-Fanelli”, Sapienza University of Rome, 00185 Rome, Italy; 3Faculty of Science, University of Sarajevo, 71000 Sarajevo, Bosnia and Herzegovina; 4Institute of Biomedical and Genetic Engineering (IBGE), Islamabad 54000, Pakistan; 5Department of Molecular Medicine, Aziz Sancar Institute of Experimental Medicine, Istanbul University, Çapa, Istanbul 34280, Turkey; 6Department of Physiology and Pharmacology “Vittorio Erspamer”, Sapienza University of Rome, 00185 Rome, Italy; 7Department of Cardiovascular, Endocrine-Metabolic Diseases, and Aging, Italian National Institute of Health, 00161 Rome, Italy

**Keywords:** ribosome, cancer, target drugs, rRNA and tRNA inhibition

## Abstract

The human ribosomes are the cellular machines that participate in protein synthesis, which is deeply affected during cancer transformation by different oncoproteins and is shown to provide cancer cell proliferation and therefore biomass. Cancer diseases are associated with an increase in ribosome biogenesis and mutation of ribosomal proteins. The ribosome represents an attractive anti-cancer therapy target and several strategies are used to identify specific drugs. Here we review the role of different drugs that may decrease ribosome biogenesis and cancer cell proliferation.

## 1. Introduction

Ribosomes are conserved ribonucleoprotein complexes. The ribosome functions as two separate subunits in all kingdoms of life. Bacterial ribosomes are composed of the 30S and 50S subunits. The 30S subunit contains 21 ribosomal proteins and a 16S rRNA, which recognizes by its sequence at the 3′-end, the Shine–Dalgarno (SD) sequence or ribosome binding site (RBS) of bacterial mRNA. The large 50S subunit consists of two rRNA, namely, 5S and 23S rRNA, and 31 ribosomal proteins; this subunit is responsible for catalyzing peptide bond formation. It has been shown that it is possible to link covalently the two subunits in a single entity [1,2]. The covalent bond between two subunits modifies the properties of the ribosomes, altering their ability to initiate and terminate translation correctly [3].

In eukaryotic cells, the two ribosomal subunits are identified as 40S (small) and 60S (large). The small subunit is composed of 18S rRNA and 33 proteins, and the large one consists of three rRNAs, namely, 5S, 5.8S, and 28S, and 49 ribosomal proteins. The Kozak consensus sequence functions as the protein translation initiation site, identifying the AUG codon, in most eukaryotic mRNA transcripts. Almost all eukaryotic translation initiates at an AUG start codon; however, recent advancements in ribosome footprint mapping have revealed that non-AUG start codons are used frequently [4].

In all organisms, it has been shown that both subunits are associated (the prokaryotic ribosome as a 70S particle and the eukaryotic ribosome as an 80S particle) during initiation, rotate during elongation, and after protein synthesis dissociate. During the translation process, it has been shown that two subunits assemble to form the mature ribosome, and in this state, ribosomes are responsible for mRNA translation.

Both subunits possess different functional sites; for example, the initiation of translation is mediated by the small subunit because it carries the decoding center (DC). The large subunit mediates catalytically the bonds between amino acids because it carries the peptidyl transferase center (PTC) [5]. It is thought that the mature ribosome contains a fixed number of components, but “specialized ribosomes” with heterogeneous compositions have recently been shown to exist [6,7]. This heterogeneity exists in ribosomal protein, which seems to control the translation of protein synthesis [8]. The human ribosome has the most advanced structure compared with bacteria or lower eukaryotes [9]; for example, the 80S contains an additional expansion segment (ES). This segment has been shown to be involved with ribosomal proteins in the selection of specific mRNA during the translation process [10,11].

## 2. Ribosome Biogenesis and Signal Transduction Pathways

The ribosome is a supramolecular ribonucleoprotein complex that functions as the heart of the translation machinery to convert mRNA into protein and is defined as the cell’s protein factory. In eukaryotic cells, the synthesis of ribosomes is a complex process involving several hundred genes. Their functions include transcription of precursor ribosomal ribonucleic acids (pre-rRNAs), processing of pre-rRNAs, assembly of ribosomal proteins (r-proteins) with pre-rRNAs, and nuclear export of the ribosomal particles [12]. Ribosome biogenesis is an essential process, and mutations of genes involved in it either cause lethality or increase susceptibility to cancer, e.g., bone marrow failure, leukemia or breast cancer [13]. This biogenesis is a temporally and spatially dynamic process requiring the coordination of different trans-acting factors at different stages along the pathway, comprising no less than 170 protein factors that modify and cleave pre-rRNAs and help to assemble and export ribosomal particles [14]. To briefly schematize the mechanism of ribosome biogenesis, three DNA-dependent RNA polymerases, ~80 ribosomal proteins (RPs), and the transient incorporation of approximately 200 non-ribosomal factors are utilized in this process. The rate-limiting step is considered to be the transcription of ribosomal DNA (rDNA) into ribosomal RNA (rRNA) by RNA polymerase I (Pol I); transcription of the rDNA itself begins when the pre-initiation complex (PIC) is assembled at the rDNA promoter. PIC formation requires the binding of at least three transcription factors: the transcription initiation factor I (TIF-I), the upstream binding factor (UBF), and the promoter selectivity factor (SL1). When UBF binds to DNA, a nucleosome-like structure is formed that recruits Pol I and multiple Pol I-associated factors, forming a multiprotein complex called the Pol I holo-complex. The active rDNA genes are transcribed into the 47S rRNA precursor (47S pre-rRNA), which upon further processing (cleavage and modification) forms the mature 18S, 5.8S, and 28S rRNAs. RNA polymerase II (Pol II), in turn, transcribes the mRNAs of the RPs, while RNA polymerase III (Pol III) transcribes the 5S rRNA in the nucleoplasm. The mature 5S rRNA and RPs are translocated to the nucleolus and assembled with the other rRNAs to form the large and small ribosomal subunits, which will subsequently form the mature ribosomes after translocation into the cytoplasm and after further modifications [15].

The nucleolus is responsible for ribosome biogenesis and is formed around nucleolar organized regions (NOR), which contain ribosomal DNA gene repeats in human cells [16,17,18]. These repeats are located in the short arms of acrocentric chromosomes [19,20]. Human ribosome biogenesis requires many components. Proteomic studies have identified up to 4500 nucleolar proteins, unlike the few hundred known in yeast [21,22,23,24,25,26]. Recently, it has been shown that 625 human nucleolar proteins, approximately 30% of them processing factors, have been shown to be involved in cancer cells, and many of them have no yeast homology [27].

The processes of ribosome biogenesis and their activity (protein synthesis) are energetically expensive for the cell. The regulation of this process must be in accordance with the environmental conditions in which the cells themselves are found and with other cellular processes (e.g., cell division and differentiation). Under a low nutrient condition, ribosome biogenesis and protein synthesis would not be favorable processes for the cell. Moreover, ribosome biogenesis and protein synthesis would be catastrophic to the cell if they initiate at the moment of cell division rather than before or after cellular division [28]. In this regard, in order to ensure cellular homeostasis, the biogenesis of the ribosome must respond rapidly to environmental stimuli or stresses (oxidative stress, DNA damage, amino acid depletion, etc.) via internal and cell surface receptors [29]. Receptors are responsible for the multiple signaling pathways between extra and intracellular compartments. The numerous signaling pathways, which intersect to control ribosome biogenesis and protein synthesis, are IL-6/MYC [30,31] and (EGFR-IGFR-TNF-α)/PI3K/AKT/mTOR [31,32,33]. The components of these pathways can represent an indirect bio-target to control the altered ribosomal biogenesis [34,35].

## 3. Protein Synthesis by Ribosomes

Protein synthesis is linked to the proliferative process in normal cells [36,37]. However, in cancer cells, the mechanism of protein synthesis is related to their metabolic requirements, and it has been seen that cancer cells express 10,000 different proteins [38], and the protein synthesis process is very complex and associated with enormous energy consumption. It has been shown that all stages of protein synthesis are dysregulated in cancer cells [39].

Oncogenic signaling in mutated receptors such as EGFR (epidermal growth factor receptor), MYC, and RAS is shown to converge on mTORC1, and in this way the first step of translation, initiation of protein synthesis, is stimulated. Initiation and elongation are two important steps of translation that are shown to occur at increased levels, and this is achieved by disorders of translation factors [40]. For example, we know that the eukaryotic initiation factor 4F complex (eIF4F) plays a pivotal role in protein synthesis, but in cancer cells this factor is deregulated. In many cancer cells (not all), ribosome biogenesis is enhanced to improve protein synthesis. Protein synthesis has for a long time been considered a possible target for anti-cancer therapy [41] (Figure 1).

## 4. Ribosome Biogenesis and Cancer Pathogenesis

The alteration that occurs at the nucleolar level and is observed in cancer cells is directly reflected in ribosome hyperproduction. The increase in ribosome biogenesis is the main trend for cancer cells, and this event is shown to be important for cell transformation and tumorigenesis, but it remains unclear why some cancer cells more than others depend on ribosome biogenesis and also protein synthesis [41,42,43,44]. Recently, it has been shown that disruption of the impaired ribosome biogenesis checkpoint (IRBC) is sufficient to elicit the DNA damage response, suggesting that the IRBC acts as a barrier against genomic instability [45]. The mutation of several ribosomal proteins in somatic cells can cause cancer. For example, mutation in RPL5/uL5, RPL10/uL16, RPS15/uS19, RPL11/uL15, and RPL22/eL22 have been described in several tumors [46]. However, the increase of ribosomal protein synthesis in cancer cells has been shown to be related to dysregulating of the three RNA polymerases, involving cancer-promoting proteins such as major oncogenic and tumor-suppressive pathways, c-Myc, mTOR, p53, pRB, and PTEN [47,48]. Ribosome biogenesis is a very complex process in which four rRNA and eighty ribosomal proteins are involved [49]. Three rRNA are produced by RNA polymerase I (Pol I); this process is very extensive, and the fourth rRNA 5S is produced by RNA Pol III, while ribosomal proteins are encoded by Pol II [49].

The first event in ribosome biogenesis is the transcription process which occurred through activation of Pol I in the nucleolus. Pol I activity has been shown to vary throughout the cell cycle [50]. Upstream binding factors (UBF) are required for the efficient transcription of ribosome genes. They stimulate and enhance the activity of Pol I and ribosome biogenesis. This factor is activated with a phosphorylation process by kinases such as casein kinase II (CKII). UBF is phosphorylated by Erk1/2 (Extracellular signal regulated kinases) and is synchronized to the cell cycle through Cdk4-CycD1 and Cdk2-CycE complex (Cdk: Cyclin-dependent kinases; Cyc: Cyclin), which are upregulated in cancer cells. The defect in p53 and pRb is an example of upregulation of Pol I and Pol III activity during ribosome biogenesis and, in this form, it supports tumor growth and development [51]. Aberration of proto-oncogenic c-Myc increases 7-methylguanylate-cap-dependent mRNA translation through the cell cycle when, during mitosis, it outcompetes the internal ribosome entry site (IRES) dependent translational of mRNA [52]. A connection between the inflammation process and ribosome biogenesis is also discovered. In fact, IL-6 (an inflammatory cytokine) stimulates, in a c-Myc-dependent manner, transcription of rRNA, and this rRNA will aggregate with ribosomal proteins to form new functional ribosomes [30].

## 5. Identification of Inhibitors That Have Target rRNA

The pre-rRNA maturation is a highly coordinated process that begins with the 90S pre-ribosome, which is also called the SSU (small subunit) processome [53,54]. The pre-rRNA undergoes an endo-nucleolytic cleavage reaction, which ends with the generation of 18S, 5.8S, and 28S rRNA, with the concomitant arrival of ribosomal proteins, which are imported from the cytoplasm to assemble with these rRNAs [55].

Pol I is involved in precursor transcription (35S), and it is known that the mature rRNAs (18S and 5.8S–25S) are embedded in noncoding 5′ and 3′ external spacers (ETSs) and internal transcribed spacers 1 and 2 (ITS1 and ITS2). During the rRNA processing steps, these spacers are accurately removed to generate the mature rRNAs; see Figure 2 for details. Enzymes, namely, endo- and exonucleases, are involved in processing steps [56], and snoRNAs are also involved both in pre-rRNA modification and processing steps [56].

The processing involves three classes of snoRNAs: box C/D, box H/ACA, and MRP. The Box C/D and box H/ACA snoRNPs drive RNA modification. In particular, the Box C/D snoRNP is composed of 60 to 200 nt and is associated with methyltransferase fibrillarin (FBL, which is NOP1 in yeast). The Box H/ACA is composed of 120 to 250 nt and is associated with dyskerin pseudouridine synthase 1 (DKC1) (known as NAP57; CBF5 in yeast) [57]. The RNase MRP (RNase mitochondrial RNA processing) is a class of its own. MRP is composed at 268 nt in humans and 340 nt in yeast; it is involved in pre-rRNA processing at site A3 in ITS1 in yeast, a function that is apparently not conserved in humans [56] (Figure 2).

In the maturation of 40S ribosome subunits, CRM1 exportin has been shown to be involved in the exporting of pre-40S-Nuc particles out of the nucleolus; the accumulation of 26S pre-rRNA in CRM1-inhibited cells is probably an effect caused by an accumulation of pre 40S-Nuc particles in the nucleoplasm. Other factors are also involved in the formation of complex pre-40S-No1, such as PNO1 and RRP12, which play a pivotal role in the stability of the pre-40S-Nuc particles. The elucidation, identification, and analysis of the alteration in the assembly and export of pre-ribosomal particles in ribosomopathies, cancer, and other diseases will be a future step [58].

The inhibition of Pol I in human AML (acute myeloid leukemia) cells leads to increased apoptotic cell death, delays the cell cycle, and induces myeloid differentiation in leukemic blasts [59]. Snail1 is a key regulator of EMT (epithelial–mesenchymal transition), but Snail1 is also recruited to the rDNA in cells undergoing EMT. Inhibition of Pol I by different pharmacological substrates has been shown to block EMT and induce tumor cell differentiation in mice [60]. Many drugs could be involved in ribosomal processing by slowing down or blocking ribosome biogenesis. This has been shown by many authors. On the NIH clinical collection website, 446 small molecules have been registered that have already been used in clinical studies (https://commonfund.nih.gov/molecularlibraries/tools) [61]. The drug diazaborine has been shown to bind with Drg1 and specifically block ATP hydrolysis, preventing Rlp24 release from pre-60S particles [62,63].

Tanshinone IIA has been shown to reduce 27SA2 pre-rRNA and 20SA2 pre-rRNA [61,64,65,66]. Megestrol acetate is involved in the reduction of 20S pre-rRNA and 27S pre-rRNA, suggesting that it might also exert its effect even before the separation of the 40S and 60S maturation pathway [61,67,68,69]. The drug berberine HCl has been shown to be a multiple inhibitor; the effect of this drug has been shown to cause a clear accumulation of precursor rRNAs and delay in the processing of A0, A1, and A2 [61,70]. Few substrates have been shown to cause significant accumulation of 20S pre-rRNA. These substrates are usnic acid, [71,72] celastrol [73], parecoxib Na [74,75], and carmofur [76,77,78].

The 5-FU (5-fluorouracil) derivate, carmofur, is very sensitive, and it is possible to cause 7S pre-rRNA deletion during processing; these data support the hypothesis that carmofur and 5-FU directly target the ribosome biogenesis pathway [61]. Both substrates are widely used as chemotherapeutic agents, although the main effect on the cell is not fully understood. The 5-FU is incorporated into RNA and interferes in various nucleotide pathways, including transcription and processing [79,80,81,82].

Another substrate, syringetine-3-glucoside, caused a strong accumulation of the total 27S signal, 27SA2, as well as 27SB precursors [61]. Vulpinic acid is another substrate that affects ribosome biogenesis. This substrate caused the accumulation of 7S pre-rRNA and small A2–A3 spacer fragments. Vulpinic acid has been shown to cause aberration of 23S RNA [61,83].

A similar effect has been demonstrated for fluphenazine 2HCl. This substrate has been shown to cause a slowdown in the early step of processing (A0, A1, and A2) [84].

Idarubicin, a member of the rubicin substrate family, can cause defects in pre-60S ribosome maturation, and accumulation of 7S pre-rRNA. Members of this family are daunorubicin, doxorubicin, and epirubicin. Doxorubicin has been reported to cause the blocking of rRNA transcription in human cells [85,86].

Two other members, doxorubicin and epirubicin, lead to a change in nucleolar morphology and nucleoplasmic accumulation of an RPL27 [61]. Rubicin interferes in pre-ribosome maturation and causes different patterns of pre-rRNA processing defects in different steps of ribosome biogenesis [61]. The members of the rubicin family block the replication process by inhibiting topoisomerase I or the possibility of intercalating in DNA strands and are widely used for the treatment of solid tumors [87,88]. Three substrates, namely, streptonigrin, acivicin, and mycophenolic acid, have been shown to cause an almost complete disappearance of pre-rRNA for a half-hour after treatment. Avicin and mycophenolic acid are known to influence the inhibition of purine and pyrimidine base synthesis. The streptonigrin has been shown to act in a completely different way. This substrate will be complexed with DNA molecules and affect the transcription and replication of DNA [61,67,68,69,70,71,72,73,74,75,76,77,78,79,80,81,82,83,84,85,86,87,88,89]. Cantharidin has been shown to induce the overexpression of several components of 3′–5′ mRNA decay in mammalian cells, including two core components of the exosome, which are clues for the connection of the drug target to the 3′–5′mRNA decay pathway [90]. Tunicamycin B is involved in the activation of unfolded proteins and has been shown to downregulate the transcription of ribosomal protein genes [91,92,93]. The substrate methotrexate blocks dihydrofolate reductase [94]; during this process, nucleotide synthesis, transcription of rRNA, and S-adenosylmethionine, which is responsible for the methylation of rRNA, are affected [85,95,96] (Table 1).

Blockage in the early step has been shown to cause nucleolar disintegration, whereas blockage in the later step during rRNA processing leaves the nucleolus intact. The drugs that we mentioned earlier will inhibit 47S rRNA precursor, but the question remains whether these drugs act directly as an inhibitor of distinct steps of ribosome biogenesis. Oxaliplatin/5-FU, and methotrexate/5-FU combination therapies are currently used in different clinics for the treatment of various types of cancers [85].

It is well known (as described) that drugs inhibit ribosomal RNA in different steps:(a)rRNA transcription: the drugs that participate in this step, for example, are oxaliplatin, doxorubicin, and methotrexate,(b)Early rRNA processing: the drugs that act in this step, for example, are berberine HCl, negestrol acetate, and tanshinone IIA,(c)Late rRNA processing: the drugs that participate in this step, for example, are 5-fluorouracil and homoharringtonine.

Protein such as c-Myc and N-Myc have been shown to induce tumorigenesis and have been the focus of research in recent decades. Due to overexpression of the c-Myc and N-Myc oncoproteins, most neuroblastoma patients die as a result of this disease. The long noncoding mRNA lncNB1 binds ribosomal protein L35/uL29 to enhance E2F1 protein synthesis, B-Myc protein stability, and N-Myc-driven oncogenesis and could be considered as a therapeutic target [97]. In this regard, the product of the gene *SMARCB1*(SWI/SNF related, matrix associated, actin dependent regulator of chromatin, subfamily b, member 1), known to encode the SNF5 subunit of the SWI/SNF chromatin remodeler, interacts with the oncoprotein transcription factor MYC and stimulates MYC activity. Weissmiller et al., (2019) [98] showed that SNF5 inhibits the DNA-binding ability of MYC. The MYC is regulated by SNF5, separately from its role in chromatin remodeling, and the reintroduction of SNF5 into the SMARCB1-null cell simulates the primary transcriptional effects of MYC inhibition. These reveal that SNF5 will antagonize MYC and provide a mechanism to explain how the loss of SNF5 can stimulate malignancy [98].

It is also known that several RPs (ribosomal proteins)are very important in miRNA-mediated modulation of the *MYC* oncogene. This control process is regulated by the transcription of rDNA, Afs (assembly factors), and RPs, and translation initiation factors [31] uL5 (RPL11), uL18 (RPL5), and uS11 (RPS14) are involved in the nucleolar stress response by stabilizing TP53 (tumor protein p53), and these factors are also accompanied by MYC transcript silencing by hsa-miR-24 [99]. RPL5 and RPL11 can bind the Myc box II domain and then inhibit its transcriptional activity through a mechanism of miRNA targeting mediated by the RNA-induced silencing complex [100,101]. RPS14 also promotes c-Myc’s mRNA turnover and decreases c-Myc transcriptional activity [102]. Investigations on c-Myc expression and activity have reported that cytosolic ribosomal proteins (CRPs), uL18 and uL5, play a role in its transcriptional activation managed by the RISC-mediated miRNA targeting mechanism [103]. It was reported that RPS2 ribosomal protein was over expressed in malignant prostate cancer cell lines and in archived tumor specimens. These data suggested that the targeting of uS5 can be a good therapeutic strategy for prostate cancer [104].

Recent studies have provided new perspectives on miRNA affecting cancer treatments. The balance between proliferation and oncogenesis can be achieved by silencing upregulated oncogenes such as *MYC, TP53*, *RPs*, and AFs downstream of pre-rRNA transcription [105]. It has been reported that hsa-miR-7641 can directly suppress (RPS16) and other RPs. In this regard, the depletion of miR-7641 sensitized the colon and also breast cancer cells to doxorubicin-induced apoptosis [106].

Identification of inhibitors for ribosome biogenesis remains a task for future study, and selective inhibitors will be valuable tools to facilitate and understand different steps.

## 6. Targeting Pol I Transcription for Therapeutic Effect

Ribosome biogenesis is a very coordinated process. This process, as we mentioned before, involves three RNA polymerases. Targeting ribosome biogenesis through inhibition of Pol I has several advantages: (1) Pol I is a highly selective process since Pol I transcribes pre-RNA; (2) ribosome biogenesis is a deregulated process in most, but not in all, cancer cells, and Pol I inhibitors have great potential to treat cancer diseases; (3) in healthy cells, the level of ribosome biogenesis is low, and this make these cells very insensitive to the effects of Pol I inhibition [107].

CX-3543 is known as quarfloxin. CX-3543 can dissociate nucleoline from putative G4 structures in the rDNA locus [108]. Nucleolin has a role in several stages of ribosome biogenesis: It facilitates Pol I transcription by promoting the euchromatic state of rDNA loci [109,110]; it catalyzes the cleavage of 5′ ETS; and it is involved in the assembly and transport of the ribosomal subunits [111]. CX-3543 inhibits rRNA transcription and leads to the stabilization of p53 and to the induction of apoptosis. Treatment with CX-3543 showed clear efficacy in the inhibition of several cancer cell lines [108]. Phase I clinical trials of CX-3543 have been completed, and now phase II clinical trials have been reached for neuroendocrine and carcinoid tumors [112,113].

CX-5461 was first identified as a selective inhibitor of Pol I transcription [114]. Inhibition of Pol I is shown to be irreversible, which is important for the design of chemotherapeutic strategies and to avoid drug resistance [115]. It has been shown to inhibit Pol I at low concentrations [116]. CX-5461 has shown a therapeutic effect in AML and prostate cancer [117,118,119], breast cancer [120], small cell lung cancer [121], ovarian cancer [122], and neuroblastoma [123]. The combination of CX-5461 with CX-6258 significantly reduces tumor volume in prostate cancer compared with a vehicle control [119].

BMH-21: has a high potential to induce p53 activation. BMH-21 prefers to bind to GC-rich DNA. This compound has been shown to inhibit rDNA transcription due to the disassembly of the Pol I complex at the rDNA promoter [124] (Table 2).

## 7. Identification of Inhibitors That Have a Target Translation Process

Drugs that inhibit bacterial growth are the most successful medicines found by humans and have saved millions of lives. Finding new natural and synthetic antibacterial drugs is one of the crucial challenges for modern health science. The knowledge of current drugs is very limited. Knowledge of how antibiotics achieve inhibition effects on their target and on cell growth is based on research carried out decades ago. To date, ribosomes are targeted by different natural or synthetic antibiotics during protein synthesis. These antibiotics bind ribosomes in different parts and lock the functional center, thereby preventing the access of tRNAs or interfering with the action of the translation factor [125]. Most antibiotics inhibit bacterial cell growth and proliferation by targeting essential cellular enzymes. These enzymes are involved in different catalytic reactions when two or more substrates combine to form a new molecule. Inhibition of these enzymes means slowing cell growth. A wide variety of clinically used antibiotics has achieved their therapeutic effects by interfering with ribosome function. In fact, ribosomes are shown to be vigorously involved in translation in fast cell growth (Figure 3).

A single mammalian cell expresses on average 105 to 106 cytoplasmic ribosomes at a given time, but this number may vary [126,127]. This pool is regulated by ribosome biogenesis and is suitable for cell needs [31,128]. The ability of the free ribosome is a very limited parameter during the translation process [129], and any quantitative changes in ribosome homeostasis can impact the translation process [130,131].

The ribosome has been seen as a biological machine dedicated to protein synthesis. This is based on the nature of its task, and its remarkable efficiency in performing it: decoding mRNA at 5.6 codons per second in eukaryotic cells [132].

Garreau de Loubresse et al., (2014) published an interesting study in which some high-resolution structures of 80S ribosomes (from *Saccharomyces cerevisiae*) were determined in complexes with 12 eukaryote-specific and 4 broad-spectrum inhibitors. All inhibitors were found associated with mRNA and tRNA binding sites. In particular, the authors suggested a model for the action of cycloheximide and lactimidomycin, explaining why the latter compound specifically targets the first elongation cycle in eukaryotic cells [133].

Peptide bond formation occurs in the peptidyl transferase center (PTC); this is the only catalytic reaction in which ribosomes are involved in the rearrangement of covalent bonds. The PTC is a prevalent target for protein synthesis inhibitors [134,135]. The chloramphenicol antibiotic (CHL), produced by *Streptomyces* species, is one of the oldest known PTC-targeting drugs [136]. CHL has been shown to bind in the aminoacyl (A) site, occupying the place of aminoacyl–tRNA [137,138,139]. In this form, CHL blocks formation of each peptide bond.

Another drug that has been shown to bind with PTC is linezolid (LZD), a synthetic oxazolidinone antibiotic introduced in the clinic 50 years after CHL [140]. Like CHL, LZD was thought to bind on PTC and inhibit peptide bond formation between amino acids [141]. Additional drug components are macrolides, which are successfully used as ribosome-targeting drugs. Erythromycin (ERY) and azithromycin (AZI) have been used for the treatment of infection diseases [142,143]. Other member of the macrolide family are ketolides, which are used but with limitations due to toxicity issues [144]. Pikromycin is a member of the ketolide family, which can arrest ribosomes less efficiently, even at the commonly difficult +X+ motif [145]. Macrolides have been shown to bind at a short distance from the PTC in the nascent peptide exit tunnel (NEPT) [138].

Kasugamycin (KSG) is another substrate that is used for inhibition of translation initiation [146], whose binding site in the 30S overlapped with the last two nucleotides of the exit (E) site codon [147,148]. KSG distorts mRNA in the ribosome and, in this form, prevents recognition of the start codon. KSG action is strongly dependent on the nature and structure of the mRNA [149]. The substrate paramycin (PAR) has the ability to inhibit protein synthesis in the living world. Like KSG, pactacmycin (PAC) has been shown to bind with the 30S subunit E site in the mRNA channel [150]. Small molecule PF06446846 has been shown to bind with ribosomes [151]. The location of the binding site is not well understood, but the mechanism of action is thought to be the same as for macrolides (Table 3).

Merafloxacin, a fluoroquinolone antibacterial, as a -1PRF (programmed -1 ribosomal frameshifting) has been shown to inhibit SARS-CoV-2. This inhibition by merafloxacin is robust to mutation with the pseudoknot region and is similarly effective on -1PRF of other beta coronaviruses. This approach represents a strategy for antiviral effects for the treatment of SARS-CoV-2 [152].

## 8. Drugs That Have a Ribosome as a Target in Cancer Diseases

An interesting study by Lamb et al., (2015) showed that five classes of mitochondrial-targeted antibiotics (including erythromycins, tetracyclines, glycylcyclines, and chloramphenicol) can be used to eradicate cancer stem cells. This approach has been linked to the evidence of a close dependence on mitochondrial biogenesis for the clonal expansion and survival of cancer stem cells [153]. Regarding this approach, a more recent review analyses the use of antibiotics to treat very different conditions (cancer, neurodegenerative or mitochondrial diseases) and their effect on mitochondria. In particular, the authors observe that the bioequivalent dose required to block cancer stem cells and the molecular pathway involved in this process should be assessed carefully, because every drug or treatment will have sometimes have disagreeable side effects [154]. Another study, conducted by Myasnikov et al., (2016) investigated the role of eukaryote-specific antibiotics and their anti-proliferative effect on several cancer cell lines at the molecular level. This study revealed the specificity of different eukaryote-specific antibiotics towards cytosolic rather than mitochondrial ribosomes, suggesting the human ribosome as a cancer target [155].

Furthermore, as proof that ribosomes are potential anti-cancer targets, it has been shown that if the gene for ribosomal protein eS6 (S6RP) is deleted, it can cause cancer cell inhibition [156]. Haplo-insufficiency of the ribosomal proteins eL24 (RPL24) or eL38 (RPL38) has been shown to prevent lymphoma induction in a transgenic EμMyc mouse model [38]. For example, the initiation factor 5A (eIF5A) of eukaryotes is modified at the post-translational level by the addition of the amino acid hypusine [157]. This modification plays a pivotal role for the ribosome to synthesize proteins with proline stretches. Modification of hypusynation of eIF5A by GC7 (inhibitor of deoxyhypusine synthase (DHS)), could be an approach for cancer therapy with a high rate of protein synthesis [158].

Many oncogenic proteins, such as c-Myc, have a short half-life (≈15 min), which is rapidly affected by ribosome inhibition, compared with other oncoproteins with a longer half-life. C-Myc is a strongly oncogenic protein in several cancer types, including acute lymphoma [159]. In this regard, we surmise that ribosome inhibitors may be highly effective in cancer diseases, which depend on such short half-life oncogenic proteins.

Homoharringtonine (HHT) is an ester of cephalotoxine, which was discovered from *Cephalotaxus harrigtonina* in 1963; the harringtonine alkaloid family includes also cephalotaxine, isoharringtonine, and harringtonine [160]. Homoharringtonine has the ability to bind 80S human ribosomes when they synthesize diphenylalanine in the presence of poly-U mRNA, elongation factor (eEF1), and Phe-tRNAPhe [161]. The inhibition of translation by HHT is much higher in eukaryotes than in archaea, so the idea that HHT is a specific inhibitor for eukaryote ribosomes is supported [162]. HHT has demonstrated an antiproliferative function on murine leukemia cells and has been approved as a drug for the treatment of chronic myeloid leukemia (CLM) patients by FDA in the USA [163]. Historically, HHT was the first compound used against ribosomes to inhibit protein synthesis during the treatment of patients with cancer [164]. HHT can be combined with other substances during the treatment of different diseases, such as in combination with oridonin on AML [165]; with SAHA, a histone deacetylase inhibitor by upregulating the expression of death receptors at the AML cells [166]; and with bortezomib to kill diffuse large B cell lymphoma (DLBCL) [167].

One study provided stronger evidence for the hypothesis that platinum-based chemotherapy (oxaliplatin) is possible to cause cell death via ribosome biogenesis [168], whereas cisplatin is shown to cause cell death via the DNA damage response (DDR). Oxaliplatin has been shown to cause inhibition of ribosome biogenesis. In particular, Sutton et al., (2021) showed that using equivalent doses of oxaliplatin inhibits ribosomal RNA synthesis by Pol I, but this does not occur with cisplatin. Redistribution of nucleophosmin (NPM1) and fibrillarin was demonstrated in the oxaliplatin-treated sample [169].

eFT508 (tomivoserbit) is a compound used for the treatment of solid tumors; this substrate has been shown to suppress protein synthesis such as oxaliplatin and 5-fluorouracil. This drug, in combination with paclitaxel, is currently used in breast cancer treatment [170,171].

Triple-negative breast cancer (TNBC) is one of the aggressive cancers for which therapy is lacking. Makhale et al., (2021) tested combination therapy using CX-5461 (selective inhibitor for Pol I transcription) and APR-246. This combination was shown to significantly induce apoptosis associated with PARP and caspase 3 along with annexin V [172].

Osteosarcoma frequently occurs in children and adolescents and cause a poor prognosis. The role of RBPs (RNA binding proteins) has been explained in recent years. Li et al., (2021) identified the key RBPs in osteosarcoma, which are the prognostic factor treatment targets. Thirty-eight differential expression RBPs were identified in this study and the results indicated that these RBPs were significantly involved in ribosome biogenesis and mRNA surveillance pathway. The genes *DDX24*, *DDX21*, *WARS*, and *IGF2BP2* might play a pivotal role in osteosarcoma, and these genes will be considered as therapeutic targets for osteosarcoma treatment [173].

Small molecules that are inhibitors of mitosis, such as the KIF11 inhibitors ispinesib, nocodazole, and paclitaxel, and the aurora kinase inhibitors hesperidin and MK-5108, decrease and inhibit DNA replication and increase the nucleolar number during mitosis [174].

JEB (junction epideromolysis bullosis) is caused by premature termination codon (PTC) mutation in skin cells, anchoring the protein LAMB3 gene (laminin subunit beta 3). It has been shown that ribosomes are responsible for most of the translation reads of LAMB3PTC mRNA, which produce non-functional protein. Modifications of RPL35/uL29 increase the production of the full length Lamb3 protein from LAMB3PTC mRNA. Atazanavir and artesunate were identified as candidate small molecules for binding with RPL35/uL29 and possibly trigger increased production of full-length Lamb3 protein from LAMB3PTC mRNA for targeted systemic therapy in treating JEB patients [175].

Esophageal squamous cell carcinoma (ESCC) is a very aggressive cancer, and the ΔNp63α/RSK4/GSK-3β axis (RSK4: ribosomal protein S6 kinase A6; GSK: glycogen synthase kinase 3β) plays a pivotal role in radioresistance in ESCC. Combination therapy involving inhibitors of RSK4 and radioresistance is a good opportunity for the treatment of patients with ESCC [176].

Tau is a neuronal-enriched microtubule-associated protein whose main function is to regulate different molecular processes, such as synaptic plasticity, cell signaling, molecular trafficking, and axonal transport [177,178,179,180]. The expression of human tau protein has been shown to decrease protein synthesis and ribosome biogenesis. Expression of the amino-terminal domain of human tau is sufficient to reduce protein and ribosome synthesis, as shown by Evan et al., (2021) [181].

Furlan et al., (2021) quantified two ribosomal proteins, RPL36 and RPL29, which are overexpressed in enzalutamide, resistant in prostate cancer, and downregulated upon BD/HAT (bromodomain/histone acetyltransferases)-inhibition treatment [182]. Other ribosomal proteins have been reported in prostate cancer pathogenesis, such as RPL19 [183], RPL21, and RPL24, and have been proposed as good prostate cancer biomarkers [184] (Table 4).

## 9. rRNA and Ribosomal Protein Modification in Cancer Diseases

rRNA carries more than 100 chemical modifications, including pseudo-uridinilation, methylation, and ribose methylation at 2′-hydroxyl [185]. During modification, the ribosome stabilizes its structure, and modifications are a cluster for important functions of the ribosome, such as the peptidyl transferase center (PTC) and decoding center, which in this form promotes accuracy and efficiency of the decoding process. The most abundant rRNA modifications are uridine to pseudo-uridine (Ψ) via enzyme pseudo-uridine synthases and H/ACA box small nucleolar RNAs (snoRNAs), and 2′-O-methylation of the ribose via enzyme methyltransferase fibrillarin (FBL) and C/D box small nucleolar RNAs [57,186]. The existence of rRNA 2′-O-methylation plasticity would control the intrinsic capabilities of ribosome to translate IRES-containing mRNA [187,188]. This means that rRNA modification according to chemical patterns may present a new strategy for creating a new type of ribosome called the specialized (therapeutic) ribosome. The role of expression and modification of rRNA is slowly emerging. It has been shown that there exists a correlation between increased expression and modification of rRNA in cancer cells [189], such as prostate and cervical cancer [190,191], and high expression of pre-45S rRNA has been shown in colorectal cancer during the G1/S cell cycle [192]. In rare genetic diseases such as X-linked Dyskeratosis Congenital (X-DC), hypermodification of rRNA has been found; the gene *DKC1*, which encodes dyskerin, is involved in the modification of pseudo-uridinilation of rRNA in approximately 100 specific sites [193]. In breast cancer, expression of FBL (fibrillarin) has been shown to alter the rRNA 2′-O-methylation pattern, triggering changes in the translation of mRNAs [187,194]. An existing link between 2′-O-methylation and pseudo-uridynilation to cell proliferation, host immunity, and oncogenic microRNAs in malignant melanoma (MM) suggests that both RNA modifications and other factors that are involved in this process are good targets for tumor therapy and good prognostic cancers [195].

Many proteomic studies have identified ribosomal protein modifications associated with different diseases. Modification of RPS6 is associated with different physiological and pathological cellular contexts. Phosphorylation of RPS6 has been shown to stimulate translation of specific class mRNA containing a 5′TOP sequence in response to mTOR signaling [196].

RPS15 has been shown to be a substrate of LRRK2 kinase (leucine-rich repeat kinase). A mutation of LRRK2 kinase leads to neurodegenerative disorder diseases such as Parkinson’s disease (PD) [196,197]. Phosphorylation of RPL12 (serine 38-pS38) was found to be abundant in both 60S and 80S fractions, but not in the polysome fraction, suggesting that phosphorylation of RPL12 in this position may regulate translation [198].

Fucosylation is another post-translation modification that contributes to pathogenesis of several diseases, although to date is unclear how and which proteins, signaling pathways, and cellular processes are implicated in fucosylation. It is thought that fucose binding lectin and many other intracellular proteins undergoes post-translation fucosylation. The ribosomal protein S3 (RPS3) is fucosylated in human cancer cells and normal mouse tissue [199].

## 10. Other Roles of Ribosomal Proteins: Modulation of the Immune System

The ribosomal proteins of both subunits specifically modulate the expression of MHC-I (Major histocompatibility complex class I) peptide cell surface expression. RPS10, RPS13, RPS28, RPLP0, and RPL3 have been shown to regulate cell surface human class I molecules including HLA-A2. The ribosomal proteins RPS7, RPS15A, RPL6, RPL17, RPL28, RPL38, RPL39, and RPL40 regulate K^b^-SIINFEKL generation without affecting viral protein translation.

The knockdown of RPS28/eS28, RPL6/eL6, and RPL28/eL28 has slight effects on the transcriptome and likely regulates class I peptide presentation through mechanisms independent of regulating the individual mRNA transcript. It has been shown that the knockdown of these proteins also potentially alters ribosome function by inducing a change in the association of ribosomal proteins and methylation (modification) of rRNA.

RPL6/eL6 and RPL28/eL28 are located near to each other in the ribosome, within several contact residues. Despite their position, they play opposite roles in modulating peptide generation. RPL6/eL6 knockdown selectively inhibits Ub-dependent peptide generation, implicating RPL6 in the ubiquitylating or degradation DRiP (rapid degradation nascent polypeptide), while RPL28/eL28 has been shown to enhance the model peptide SIIFEKL (TAP/ubiquitin/proteasome dependent/independent) [200].

## 11. The Ribosome Inactivation Proteins as Anticancer Therapy

RIPs (ribosome inactivation proteins) are a group of cytotoxic N-glucosidases. A large number of them come from plants, and a few come from bacteria [201]. RIPs are classified into three different types. Type one, trichosanthin (TCS) and momorcharin (MMC), consists of a single chain with catalytic activity. Type two, heterodimeric ricin and abrin, consists of two chains (A and B) connected with disulfide bonds and with an active A chain. Members of type three are maize ribosome-inactivating protein and barley jasmonate-induced RIP (JIP60) [202,203].

RIPs have the possibility to remove a specific adenine in the α-SRL (α-sacrin/ricin loop) of rRNA. This is a highly conserved loop in all large ribosomal subunits and is essential for the correct assembly of the functional core of this subunit and for the (GTP)-dependent binding of elongation factors to the ribosome [204].

TCS (trichosanthin) has anti-tumor activity in a wide spectrum of cancers. TCS acts as an inhibitor of cervical cancer cells through restriction of the signal transducer and activator of transcription (STAT5)/c-Myc signaling pathway. In B-cell lymphoma, expression of antigen ki-67-associated cell proliferation and RNA transcription is decreased, while caspase-3 activity is increased [205]. TCS has been shown to mediate the phosphoinositide 3-kinase (PI3K)/protein kinase (AKT) pathway and enhance the cytotoxicity and apoptosis-inducing activity of gemcitabine in non-small cell lung cancer [206]. TCS also enhances the cell uptake of granzyme B, leading to apoptosis of tumor cells [207]. TCS was able to inhibit angiogenesis in JAR cells, decrease VGFR (vascular growth factor receptor), and contribute to the anti-cancer effect [208]. TCS downregulates the NOTCH signal in the nasopharyngeal carcinoma (NPC) cell line CNE2 [209].

Ricin exhibits anti-tumor properties, and it has been shown to inhibit the growth of sarcoma in rat [210] and increase the survival rate of Ehrlich ascites tumor-bearing mice [211]. Confirmation of its properties has come from a phase I clinical study on cancer patients with different tumors. The inhibition of protein synthesis was considered the first attribution of its anti-cancer activity [212]. Ricin has been shown to induce the production of anti-inflammatory cytokines such as tumor necrosis factor alpha (TNF-α) and interleukin-1beta (IL-1β) [213,214].

A type-two RIP, riproximin, up-regulates the anti-cancer cytokine IL24/MDA-7 and ER-stress-related GADD in human and rat colorectal cancer (CLC) cell lines [215]. It also has an anti-apoptotic (BCL family), and cell cycle (cyclins) control activity in human breast cancer cell lines, such as MDA-MB-231 and MCF-7 [216].

α-MMC (lpha-momorcharin) has been shown to have an anti-cancer effect and has been tested in human breast cancer cell line MDA-MB-231 and MCF-7, but α-MMC has been shown to have high cytotoxicity and for this reason has limited use [217]. α-MMC will inhibit the immune system through the inhibition of different cytokines such as IL-1β, IL-2, IL-8, IL-9, IL-12, MIP-1α/β, and MCP-1 [218].

Another member of the RIP family is curcin, which can inhibit the growth of several tumors such as osteosarcoma cell line U20S [219].

Articularin-D (RIP family member) can selectively inhibit different cell lines such as T-cell leukemia [220].

## 12. Future Perspective and Conclusions

The key point of this review is to suggest drugs that have as a target the ribosome in cancer cells, as opposed to normal cells, because in cancer cells the protein synthesis is unbalanced, and many pathways are shown to have an influence. Combination therapy between many drugs has proven to be very effective in various cancer cells, but when it comes to inhibiting ribosome biogenesis with classic chemotherapeutic drugs, targeted therapies, or immunotherapies, a greater lethal effect can be achieved than with current treatment. The cancer cells might express onco-ribosomes that are different from normal ribosomes and participate in the process of cell transformation. The ribosomes from normal cells are heterogeneous and specialized. Several ribosomal proteins, such as RPS7/eS7, RPS25/eS25, and RPL10/uL1, have been shown to be in sub-stoichiometric abundance and demonstrate different roles [11].

Pol I transcription can become a novel therapeutic approach in the fight against cancer diseases. It has been convincingly demonstrated that many anti-cancer drugs can target various steps of ribosome biogenesis and rRNA synthesis [85]. Inhibition of Pol I transcription has been shown to have an advantage in cancer treatment. Regardless of the heterogeneity in cancer cells, in most of them, ribosome biogenesis is increased, and rRNA synthesis supports uncontrolled proliferation. This makes Pol I an exceptional target in almost all cancer cells. All this makes Pol I inhibition a good indicator for therapy. Furthermore, a combination of Pol I inhibition and drugs that are used for this reason will not only increase the effectiveness of the treatment but also reduce the possibility of developing acquired resistance.

## Figures and Tables

**Figure 1 biomedicines-10-02088-f001:**
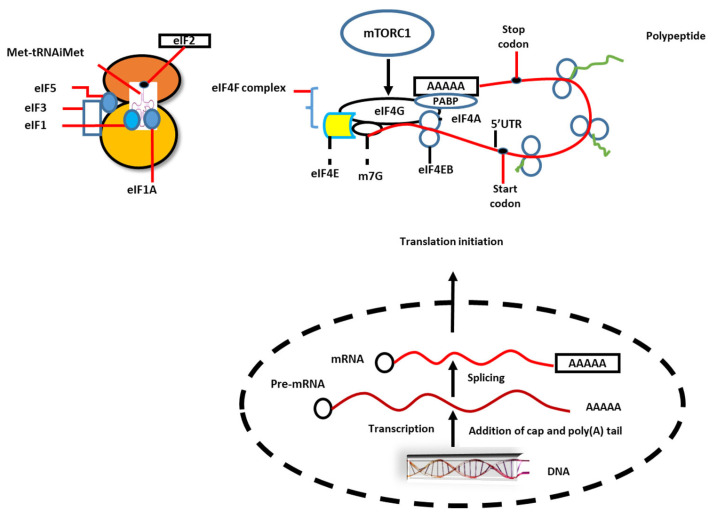
Eukaryotic mRNA undergoes several steps of processing in the nucleolus, such as 7-methylguanosine (m7G) at the 5′ and poly A-tail in the 3′-end. Ribosomes are recruited by mRNA through coordinated multiple processes. Two protein complexes, eukaryote translation initiation factor (eIF4F), which comprises eIF4E (cap-binding protein), eIF4G (scaffold protein), and eIF4A (RNA helicase), and the ternary complex, which includes eIF2-GTP and initiation tRNA (Met-tRNAiMet), have pivotal roles in translation initiation. The mRNA circularization occurs in the interaction of eIF4G with poly A-tail binding protein (PABP). The eIF4F complex displays a secondary structure in the 5′ untranslated region (5′UTR) of mRNA. The mTOR complex 1 controls the initiation of translation through the ternary complex and eIF4F complex. Moreover, the interaction of eIF4B with eIF4A increases the helicase activity of the latter.

**Figure 2 biomedicines-10-02088-f002:**
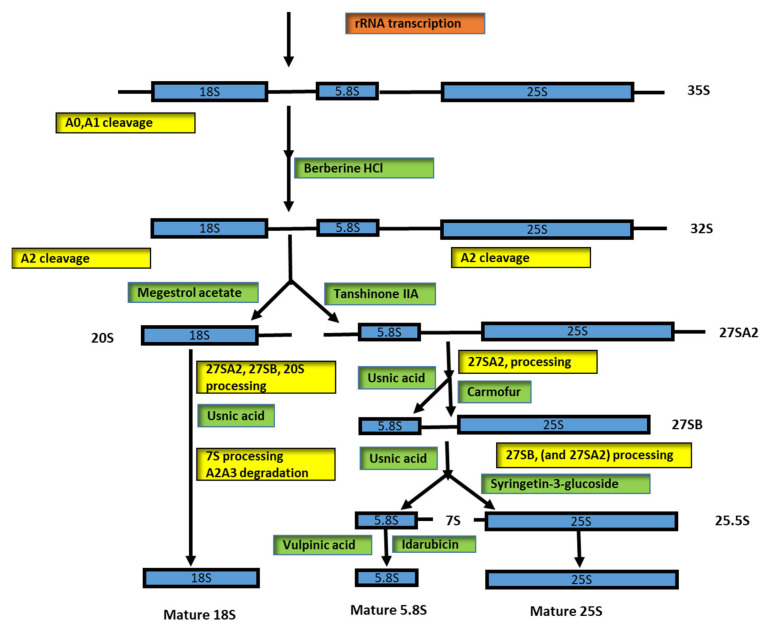
Inhibition of eukaryotic rRNA processing in different steps. The rRNA processing scheme presented here from 35S pre-rRNA to the mature rRNA (18S, 5.8S, and 25S) is complemented with different inhibitors and their potentially targeted ribosomal maturation.

**Figure 3 biomedicines-10-02088-f003:**
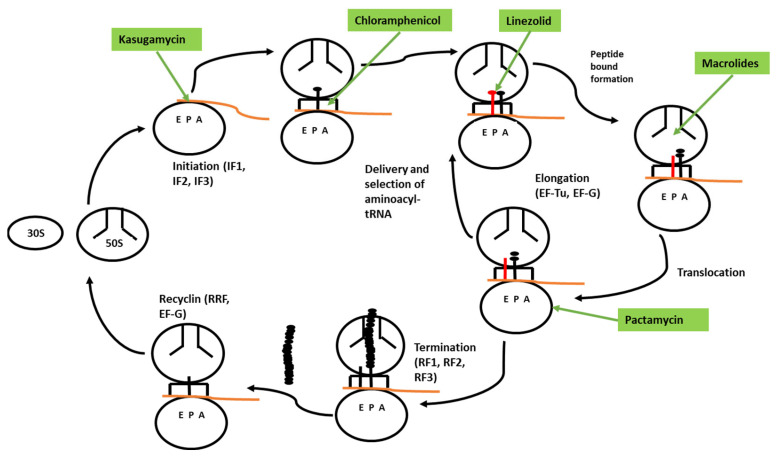
Overview of protein synthesis in bacteria and inhibition by different antibiotics. First step: initiation of protein synthesis assisted by initiation factors (IF1, 2, 3), location of the start codon in mRNA, and connection of the initiation tRNA in the peptidyl (P) site. During translation elongation, EF-Tu-delivered aminoacyl-tRNA are selected and then accommodated in the aminoacyl tRNA. The antibiotic (green highlighted)- and macrolide-inhibition of peptide bond formation depends on the structure of the nascent protein. During translocation, the A site-bound peptidyl-tRNA moves into the P site, and the tRNA with a free 3′ end is relocated into the exit E site. When the ribosome encounters a stop codon, it enters the termination phase. During this phase, the completed protein is released with the help of termination factors (RF1 or RF2 and RF3). The last step is the recycling phase, when the combination of ribosome recycling factor (RRF) and EF-G splits the ribosome into its subunits.

**Table 1 biomedicines-10-02088-t001:** Effect of different drugs on inhibition of rRNA in cancer cells.

Type of Substance	Inhibition	Diseases	References
Tanshinone IIA	27SA2 pre-rRNA; 20SA2-pre-rRNA	Cancer	[61,64,65,66]
Diazaborine	Drg1 and block ATP hydrolysis		[62,63]
Megestrol acetate	20S pre-rRNA; 27S pre-rRNA		[61,67,68,69]
Usnic acid	20S pre-rRNA		[71,72]
Celastrol	27S pre-rRNA		[73]
Parecoxib Na	27S pre-rRNA		[74,75]
Carmofur	27S pre-rRNA		[76,77,78]
5-FU (5-fluorouracil)	7S rRNA		[61]
Syringetine-3-glucoside	27S, 27SA2, 27SB precursors		[61]
Vulpinic acid	7S pre-rRNA, small fragments A2-A3, 23S rRNA		[61,83]
Fluphenazine 2HCl	Small fragments A0, A1, A2		[84]
Idarubicin	60S pre-ribosome maturation; 7S pre-rRNA accumulation		[85,86]
Doxorubicin/epirubicin	Defects in pre-rRNA maturation	Solid tumors	[61,87,88]
Streptonigrin	Transcription and replication of DNA		[61,89]
Cantharidin	3′-5′ mRNA decay pathway		[90]
Tunicamycin B	Ribosomal protein gene		[91,92,93]
Methotrexate	Transcription of rRNA		[85,95,96]

**Table 2 biomedicines-10-02088-t002:** Effect of different drugs on the inhibition of Pol I in cancer cells.

Type of Substance	Inhibition	Diseases	References
CX-3543	Facilitate Pol I transcription	Neuroendocrine and carcinoid tumors	[112,113]
CX-5461	Inhibit selectively Pol I	Prostate cancer; breast cancer; small lung cancer; ovarian cancer; neuroblastoma	[117,118,119,120,121,122,123]
CX-5461/CX-6258	Pol I	Prostate cancer	[119]
BMH-21	Pol I	Cancer	[124]

**Table 3 biomedicines-10-02088-t003:** Effect of different drugs that inhibit the translation process.

Type of Substrate	Inhibition	References
Chloramphenicol (CHL)	Peptide bond formation	[138,139]
Linezolid (LZD)	Peptide bond formation	[140,141]
Erythromycin (ERY)	Peptide bond formation	[142,143]
Azithromycin (AZA)	Peptide bond formation	[142,143]
Pikromycin	Arrest ribosome maturation	[145]
Kasugamycin (KSG)	Bind 30S subunit	[147,148]
Paramycin (PAR)	Protein synthesis	[150]
Pactamycin (PAC)	Bind 30S subunit	[150]
Small molecule PF06446846	Not known	[151]

**Table 4 biomedicines-10-02088-t004:** Effect of different drugs on inhibition of ribosome biogenesis in cancer cells.

Type of Substance	Inhibition	Diseases	References
Harringtonine	80S	Murine leukemia cells; chronic myeloid leukemia	[164]
Bortezomib/ Harringtonine		Diffuse large B cell lymphoma (DLBCL)	[167]
Oxaliplatin	Cell growth via ribosome biogenesis	Tumor/cancer	[168,169]
Cisplatine	Ribosome biogenesis	Tumor/cancer	[169]
eFT508 (tomivoserbit)	Suppress protein synthesis	Solid tumors	[170,171]
5-Fluorouracil/paclitaxel	Suppress protein synthesis	Breast cancer	[170,171]
Cx-5461/APR-246	Induce apoptosis	Triple-negative breast cancer (TNBC)	[172]
Ispinesib, Nocodazole, Paclitaxel, Aurora kinase inhibitor, Hesperidin, MK-5108	Inhibit DNA replication	Cancers	[174]
Atazanavir/artesunate	LAMB3PTC mRNA	Junction epidermolysis bullosa	[175]
Tau protein	Decrease protein synthesis	Cell signaling and axonal transport	[181]
Enzalutamide	Down regulate BD/HAT	Prostate cancer	[182]

## Data Availability

Not applicable.

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
