# Peer review of "Ribosome-Directed Therapies in Cancer"

_biomedicines, 2022, doi:10.3390/biomedicines10092088_

Round 1

Reviewer 1 Report

In their manuscript, Temaj et al review the current knowledge on potential anti-cancer therapies directed at the ribosome. The authors briefly describe the composition of the ribosomes from bacterial or eucaryotic origin, the assembly of the last one and then focus on their implication in cancer cells, their modifications and the different inhibitors that act at different levels.

Are presented inhibitors that target rRNA, ribosome biogenesis, translation function. The review is interesting to read. I have several concerns that are listed below:

Page 3, lanes 108-112. I agree that MYC is a major regulator of ribosome biogenesis, but why putting an accent on the IL6 pathway? I think there are many other external signals that stimulates ribosome biogenesis as well. What would be peculiar with the action of IL6?

I can understand that it is difficult to be original when describing the ribosome, which has been done in numerous articles and reviews.

But these two sentences:

In normal cells, protein synthesis (PS) is tightly linked to their proliferative needs

In contrast, cancer cells have enslaved protein synthesis mechanisms to fuel their metabolic needs and a typical cancer cell expresses at least 10,000 different proteins

are found in Gilles et al, Cells 2020, doi: 10.3390/cells9030629

which is even not cited in this review. This is quite unfair.

Be careful when you get inspiration from previous work to avoid plagiarism. I guess there are tools to check on that.

Paragraph 8 Drugs targeting ribosomes. I think it could be interesting to cite the work of Myasnikov et al, 2016 Nature Communications doi: 10.1038/ncomms12856.

From that, the authors could briefly discuss the respective advantages and limitations of using of eukaryote-specific and procaryote-specific ribosome inhibitors.

In this line it could be interesting for the readers to have a short discussion on the work of Lamb from Lisanti’s group that claim to support the use of bacterial antibiotics to kill mitochondria in cancer cells and eliminate cancer stem cells.

Paragraph lane 433. I think it could be worth to cite the article by Bruno et al in Nature Medicine 2017 (doi:10.1038/nm.4291) which showed that oxaliplatin acts via ribosome biogenesis stress, as this point is presented by the authors.

Figure 3: don’t you think it would be more relevant for the review to depict an overview of protein translation in eucaryote/cancer cells? The figure can also mention the use of bacterial antibiotics with the indication if they can target the 80S ribosome.

A short paragraph/couple of sentences could be added to explain how cancer cells are killed after ribosome inhibition.

Minor points:

Please thoroughly check the English throughout the manuscript. Some exemples:

Do not use who for proteins or drugs.

These titles have no sense and need grammatical correction I think:

Lane 113: protein synthesis by ribosome. Must be either the ribosome or ribosomes

Lane 172: 5. Identification of inhibitors who have target rRNA.

Lane 336 7. Identification of inhibitors who have a target translation process

Lane 403: same for the title

Lane 349: the ribosome are involved in translation, not only in the fast cell growth

Lane 187: The Pol I polymerase or Pol I, not the Pol I

Lane 333: èdue

In the introduction, it could have been nice to have the size of the whole bacterial ribosome (70S) and of the eukaryote one (80S).

Figure 2 it is not immediately obvious that the green boxes represent the inhibitors. May be you could draw links to show where they act.

Reviewer 2 Report

The review “Ribosome-Directed Therapies in Cancer” from Temaj et al. aims to give an overview of drugs that can be or are used as cancer therapeutics and target directly or indirectly protein synthesis. 

It starts with a brief introduction about ribosomes and the mechanism of translation. This is followed by the description of ribosome biogenesis and the signal transduction pathways biogenesis depends on, followed by an explanation why inhibiting translation can be used as a cancer therapy followed by a longer description of how translation and ribosome biogenesis are dysregulated/upregulated in cancer. Different drugs that act during biogenesis either against rRNA/ribosomal subunit maturation or RNA polymerase I activity are presented. Then antibiotics targeting the bacterial ribosome and are used against infectious diseases as well as some drugs targeting the eukaryotic ribosome are presented, but also the combination therapy with anti-cancer drugs that act during other cellular processes in cancer, such as microtubules/mitosis (paclitaxel). The change of chemical modifications of the rRNA during cancer and also and ribosomal protein content are elaborated as well as the modulation of downstream processes such as immune response.  Finally also the anti-tumor activity of  ribosome inactivation proteins is summarized. 

In my eyes the topic of the review is very important and should be published, however the manuscript requires thorough rewriting of some parts (especially the introduction, which should be more straight forward and less detailed, maybe some of the references could be removed for the sake of clarity, e.g. the fact the the ribosomal subunits can be linked) and a detailed language check (in some places the meaning gets lost in translation). A description of the basic steps of protein synthesis is missing in the introduction, which seeing the later focus of the manuscript should also include a basic overview of ribosome biogenesis. 

I have some further comments that could help improving the manuscript: 

-       Maybe the text could also gain some clarity if it was reshuffled and more structured (e.g. lines 260-266 could be placed earlier, since such listings aid the reader to understand the goal of the explanation). 

-       In the paragraph discussing translation inhibitors other works such as the one from Garreau de Loubresse et al. (Nature, 2014)  and Natchiar et al (Nature, 2017) could be added, which presents the binding pockets of many inhibitors targeting the eukaryotic ribosome.  

-       It should be clarified why inhibitors targeting the bacterial and the eukaryotic ribosome are discussed. Maybe the sense of these two chapters could be improved.    

-       Paragraph 10 (lines 526-541) could be either removed or discussed earlier to gain clarity. 

Specific points: 

Line 212: it would be good to give the link to the data base

Line 369: Some details presented such as the synthesis rate of the ribosome should be checked, since it is different for eukaryotic vs. prokaryotic ribosomes. 

Figure and Table presentation: 

 Figure 1 is a bit simple and could be more detailed for the protein synthesis part.

 Figure 3 could be more detailed with arrows showing where exactly the inhibitors bind. In fact also inhibitors targeting the eukaryotic ribosome could be added here, if the translation factors would be omitted.

The presentation of Tables seem good. 

Round 2

Reviewer 1 Report

Thank you for answering my comments.

I feel that this manuscript still requires some English editing. At least, it should be corrected by a native english-speaking person.

Author Response

Point-by-point response to the reviewer's comments
Comments and Suggestions for Authors (Reviewer 1-Round2)

I feel that this manuscript still requires some English editing. At least, it should be corrected by a native English-speaking person.

Author's response: We are grateful to the reviewer for his comment. We appreciate the reviewer's feedback. We have improved the manuscript with extensive revision in English.